# Post-FEC Performance of Pilot-Aided Carrier Phase Estimation over Cycle Slip

**Yan Li \*, Quanyan Ning, Lei Yue, Honghang Zhou, Chao Gao, Yuyang Liu, Jifang Qiu, Wei Li, Xiaobin Hong and Jian Wu \***

The State Key Laboratory of Information Photonics and Optical Communications, Beijing University of Posts and Telecommunications, Beijing 100876, China
\* Correspondence: liyan1980@bupt.edu.cn (Y.L.); jianwu@bupt.edu.cn (J.W.)

**Abstract:** The POST-forward error correction (FEC) bit error rate (BER) performance and the cycle-slip (CS) probability of the carrier phase estimation (CPE) scheme based on Viterbi–Viterbi phase estimation (VVPE) algorithm and the VV cascaded by pilot-aided-phase-unwrap (PAPU) algorithm have been experimentally investigated in a 56 Gbit/s quadrature phase-shift keying (QPSK) coherent communication system. Experimental results show that, with 0.78% pilot overhead, the VVPE + PAPU scheme greatly improves the POST-FEC performance degraded by continuous CS, maintaining a low CS probability with less influence of filter length. Comparing with the VVPE scheme, the VVPE + PAPU scheme can respectively obtain about 3.1 dB, 1.3 dB, 0.6 dB PRE-FEC optical signal noise ratio (OSNR) gains at PRE-BER of $1.8 \times 10^{-2}$. Meanwhile, the VVPE + PAPU scheme respectively achieves about 3 dB, 1 dB, and 0.5 dB POST-FEC OSNR gain and improves the FEC limit from $2.5 \times 10^{-3}$ to $1.4 \times 10^{-2}$, from $8.9 \times 10^{-3}$ to $1.8 \times 10^{-2}$, and from $1.6 \times 10^{-2}$ to $1.9 \times 10^{-2}$ under the CPE filter length of 8, 16, and 20.

**Keywords:** coherent communication; quadrature phase-shift keying; carrier phase estimation; cycle-slip; pilot-aided-phase-unwrap; low-density parity-check (LDPC)

---

## 1. Introduction

Recently, thanks to the advanced technology such as high-speed analog-to-digital converter (ADC), integrated optical front end, and digital signal processing (DSP), 100 Gb/s coherent single carrier transmission system has become a reality [1]. Single carrier quadrature phase-shift keying (QPSK) modulation and narrow line-width laser are a key technology in 100 Gbit/s digital coherent communication system. The laser line-width between 100 k and 500 k is commonly used. Carrier phase estimation (CPE) is an important integral part of digital signal processing (DSP) in coherent transmission systems through which laser phase noise and nonlinear phase noise are compensated [2,3]. Usually, CPE is implemented as a feed-forward structure for efficient hardware implementation, i.e., Viterbi–Viterbi phase estimation (VVPE) [4]. Due to the necessary phase unwrapping algorithm in these estimators, cycle slip (CS) occurs, which can lead to continuous errors that cannot be corrected properly by the forward error correction (FEC) decoder. A common approach to deal with CS is to use differential encoding (DE), however, differential decoding leads to error duplication and introduces penalties to the PRE- forward error correction (FEC) bit error rate (BER), thus leading to degraded POST-FEC BER [5].

A great deal of efforts has been conducted to mitigate or reduce the impact of CS and to improve the PRE-FEC BER and POST-FEC BER. A lot of research works focus on improved CPE algorithms, which can reduce the CS probability and decrease the PRE-FEC BER. For example, joint polarization carrier phase estimation (JP-CPE) [6,7], pilot-symbols-based CPE (P-CPE) [8], a forward and backward

(FWBW) method [9,10], recursive probability-weighted blind phase search (RW-BPS) [11], filtered carrier-phase estimation (F-CPE) [12], a support vector machine (SVM)-based boundary creation [13], and a signal recovery method by employing soft decision slip state estimation method at pilots [14]. However, most of these improved CPE algorithms mainly focus on the performance of PRE-FEC BER. The POST-FEC performance of the improved CPE algorithms in combination with a concatenated FEC coding scheme should be taken into consideration since CS might deteriorate the performance of FEC [5]. Some other works concentrate on improved FEC encoding/decoding methods, which can reduce differential encoding (DE)-penalty or improve robustness against CS. For example, turbo differential decoding (TDD) and some improved TDD algorithms have been proposed to eliminate error floor for frequent CS [15–18]. In References [19–21], novel code designs for BCH codes were proposed and numerically confirmed, which are robust to CS. Cao et al. [22] proposed a phase noise-aware log-likelihood ratio (LLR) calculation based on a Tikhonov model. An alternative LLR calculation method, employing linear or bilinear transform [23], was also investigated to improve robustness against residual phase noise. Schmalen [20] proposed a new low-density parity-check (LDPC) code structure, which is robust to CS. Clearly, these techniques, which treated CS within FEC coder/decoder block, can improve POST-FEC BER performance effectively. However, most of these techniques suffer from increased processing complexity.

In our previous work [24], the pilot-symbols-aided phase unwrapping (PAPU), which utilizes the time-division multiplexed pilot symbols that are transmitted with data, was proposed to do CS detection and correction with the CPE in QPSK modulation. Recently, it has been employed to high-order quadrature amplitude modulation (QAM) over 1500 km standard single-mode fiber transmission [25]. In this paper, since LDPC is one of the most commonly used FEC code [26], we experimentally investigate the performance of the CPE + PAPU scheme combined with a concatenated soft decision (SD)-LDPC coding in a 56 Gbit/s quadrature phase-shift keying (QPSK) coherent communication system with laser line-width of 300 kHz. Experimental results show that, compared with CPE, CPE + PAPU can release the required PRE-FEC optical signal noise ratio (OSNR) about 3.1 dB, 1.3 dB, and 0.6 dB at BER of $1.8 \times 10^{-2}$ under the CPE filter length of 8, 16, and 20. Meanwhile, the CPE + PAPU scheme achieves respectively about 3 dB, 1 dB, and 0.5 dB POST-FEC OSNR gain compared with the CPE scheme under the CPE filter length of 8, 16, and 20. The results also show that the CPE + PAPU scheme respectively improve the FEC limit from $2.5 \times 10^{-3}$, $8.9 \times 10^{-3}$, $1.6 \times 10^{-2}$ to $1.4 \times 10^{-2}$, $1.8 \times 10^{-2}$, $1.9 \times 10^{-2}$ compared with CPE, with the CPE filter length of 8, 16, and 20.

## 2. Experimental Setup and Principles

A 28 Gbaud single-polarization (SP)-QPSK experiment has been conducted to investigate the POST-FEC performance and the CS probability of the previous proposed CPE + PAPU scheme [27]. Figure 1 illustrates the block diagram of the experimental setup. A DFB laser with 300-kHz line-width, centered at 1550 nm, is used as the transmitter laser. The independent pseudo-random binary sequences (PRBS) are encoded in sequence by outer encoder, Reed Solomon (RS) (255,239), and inner encoder LDPC. Inner encoder uses the most commonly used soft-decision code, DVB-S2 (LDPC), with 11.1% overhead [26] and its maximum decoder iteration number is 50. Pilot symbols with known information of 0.78% [27] overhead are inserted periodically per 127 symbols at the transmitter side and the phase estimated by the pilot symbols are used as a reliable reference in the CPE phase unwrap process at the receiver side. Pilot inserted data are loaded to pulse pattern generator (PPG) and then modulated by a double balanced modulator (IQ modulator) to generate 56 Gbit/s QPSK signals. An erbium-doped optical fiber amplifier (EDFA) cascaded by a variable optical attenuator (VOA) are used as noise source. An optical coupler (OC) with coupling ratio of 1:99 splits the output optical signals. One of branch is sent to the coherent receiver for signal processing, the other is used to detect optical signal noise ratio (OSNR). Agilent optical modulation analyzer with sampling rate of 80 GSa/s and bandwidth of 33 GHz is used as coherent receiver and off-line DSP. The key physical parameters in the experimental setup are specified and summarized in Table 1.

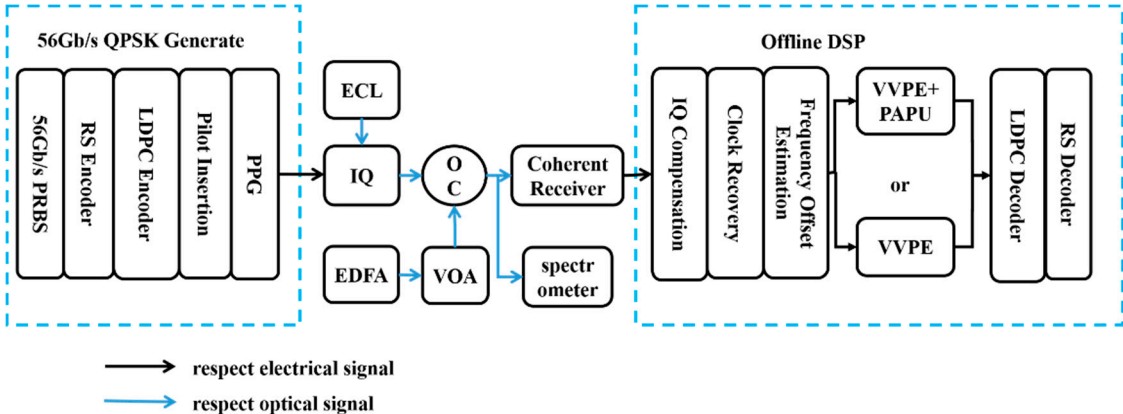

**Figure 1.** Experimental setup.

**Table 1.** Summary of the key physical parameters in experimental setup.

| Parameters | Values |
| --- | --- |
| Symbol Rate | 28-Gbaud |
| Wavelength | 1550-nm |
| Linewidth | 300-kHz |
| Bandwidth of Modulator | 40-GHz |
| Bandwidth of Receiver | 33-GHz |
| Sampling Rate | 80-GSa/s |

In the offline DSP, the signals are firstly processed by Gram–Schmidt orthogonalization procedure (GSOP) [28] to compensate in-phase (I) and quadrature (Q) imbalance. Then, the Gardner algorithm [29] is adopted to recovery clock and M-power frequency offset estimation (FOE) algorithm [30] is used to compensate frequency offset. Following the FOE, two different algorithms are employed to estimate the carrier phase. The first scenario is the traditional CPE based on VVPE, and the CPE + PAPU is adopted as the second scenario. The principle of the CPE scheme and the CPE + PAPU scheme are respectively shown in Figure 2a,b. The key point of the CPE + PAPU scheme, as shown in Figure 2b, is the pilot-symbols-aided phase unwrapping (PAPU), which does CS detection and correction in a completely forward way and combine itself with the usual phase unwrapping. Without loss of generality, we use VVPE as the CPE, and PAPU is implemented as the following procedure. After carrier phase recovery, LDPC decoder and RS decoder are conducted. Finally, the BER is calculated using 10 M bits data.

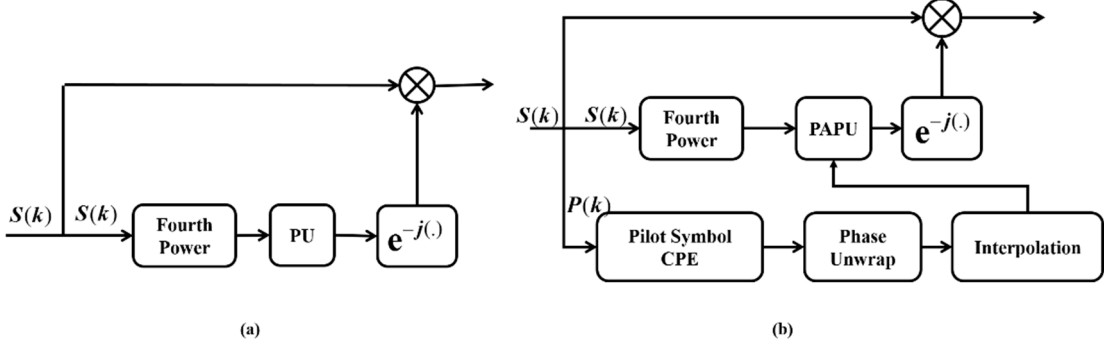

**Figure 2.** Fourth-power carrier phase recovery with (**a**) usual-phase unwrap, (**b**) pilot-symbol-aided phase unwrap.

## 3. Experimental Results and Discussion

Figure 3a–c show phase errors between the phase values estimated by CPE and the referenced phase values. Errors between CPE + PAPU and the referenced phase values are shown in Figure 3e–f. The traces in each figure indicate the errors under different CPE filter lengths (CPE filter length = 4, 8, 16, 20, 32, 48). To obtain the referenced phase values, we firstly calculate the phase error between original symbol and received symbol, then usual PU (2 pi) and an appropriate interpolation filter are used to remove the residual noise. The outputs of interpolation filter are used as referenced phase values. As shown in Figure 3, frequent CS is clearly observed for the VVPE scheme, producing ±90° phase error jumps under the condition of lower OSNR (which is equivalent to 10.5 dB or 12.1 dB) and small CPE filter length (which is less than 20). Even if OSNR is equivalent to 15.4 dB, the CPE scheme still shows frequent CS when CPE filter length is 4. Conversely, even when OSNR is equivalent to 10.5 dB and CPE filter length is less than 8, the VVPE + PAPU scheme just shows a few discrete CS, which are induced by additive white Gaussian noise (AWGN) and can be corrected by FEC. The results indicate that the CPE scheme seems to be more sensitive to CPE filter length and OSNR, compared to the CPE + PAPU scheme.

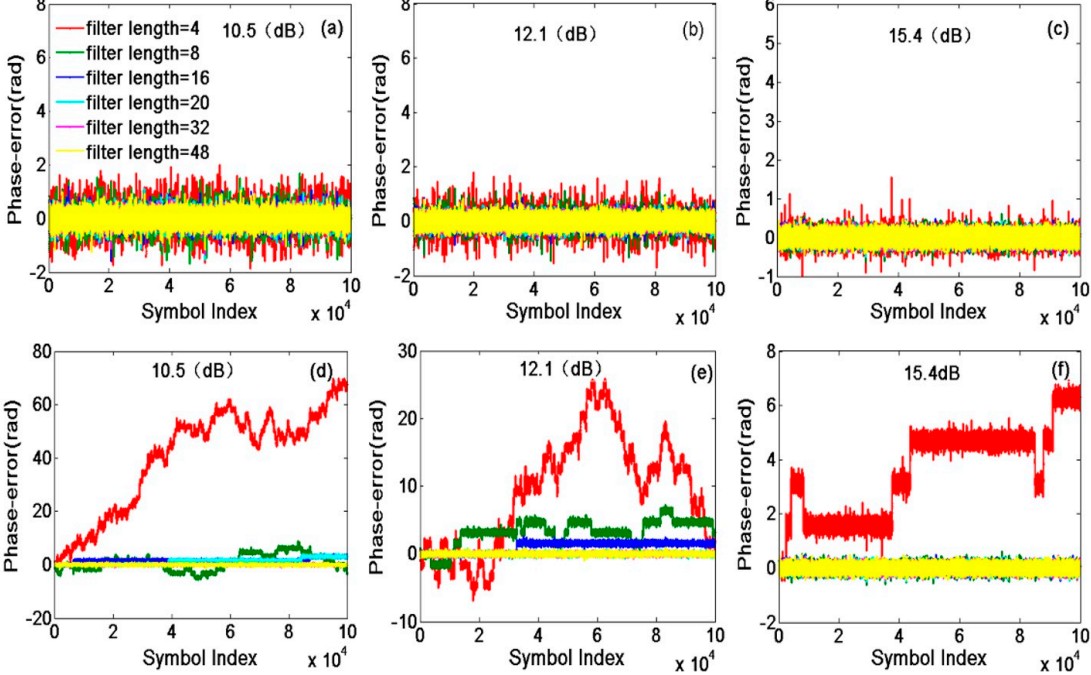

**Figure 3.** Phase errors for (**a**–**c**) the carrier phase estimation (CPE) scheme, (**e**–**f**) the CPE + pilot-aided-phase-unwrap (PAPU) scheme.

Probability of cycle slips as a function of OSNR under different CPE filter lengths are shown in Figure 4a. With the increasing of OSNR, the cycle slips probability of both schemes is decreasing and the difference between the VVPE scheme and the VVPE + PAPU scheme is becoming less obvious. The probability of CS is less than $1 \times 10^{-7}$ when OSNR is large than 17.5 dB, which indicates that the CS problem can be avoided under high OSNR. However, with OSNR less than 12 dB, the VVPE scheme exhibits a high CS-rate with CPE filter length less than 48, compared with the VVPE + PAPU scheme. The reason is that frequent continuous CS occurs in the CPE scheme with lower OSNR, while only some discrete CS exists in the CPE + PAPU scheme. The results in Figure 4a indicate that the CPE + PAPU scheme can effectively mitigate the continuous CS due to the improper PU of the CPE. The discrete CS on account of AWGN cannot be corrected by the CPE + PAPU scheme, but these independent CS can be effectively corrected by FEC. Figure 4b shows the relationship between PRE-FEC BER and OSNR under different CPE filter lengths for the CPE scheme and the CPE + PAPU scheme. The AWGN

PRE-FEC BER is presented by gray circle. There is a similar trend between Figure 4a,b. When the CPE filter length is less than 48, the sudden degrade of PRE-FEC BER is consistent with the probability of CS for the CPE scheme.

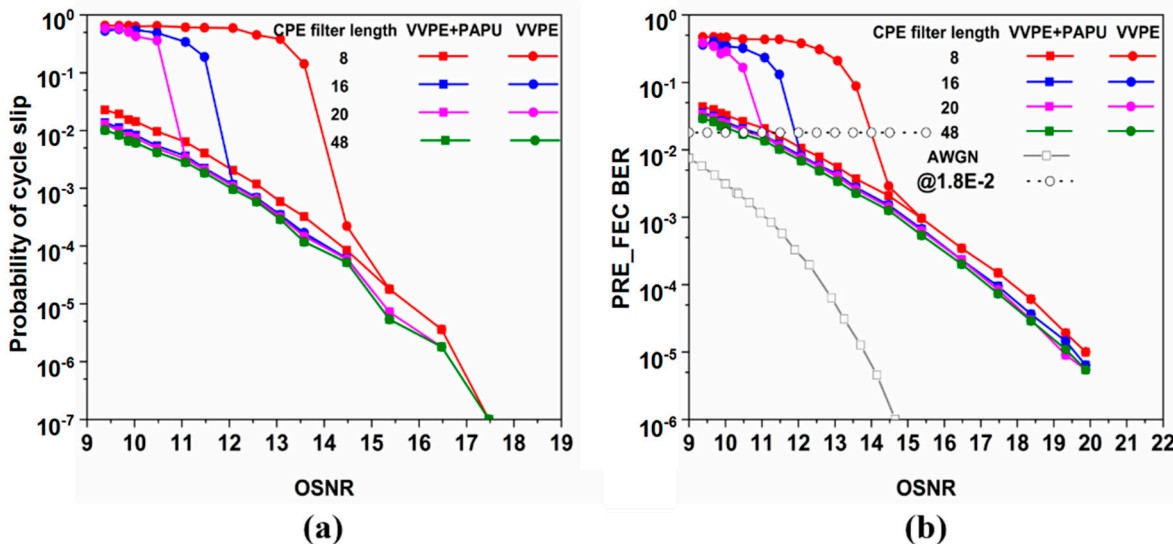

**Figure 4.** (**a**) Probability of cycle slips versus optical signal noise ratio (OSNR) for the CPE scheme and the CPE + PAPU scheme; (**b**) PRE-forward error correction (FEC) bit error rate (BER) versus OSNR for the CPE scheme and the CPE + PAPU scheme.

Figure 5a depicts the probability of CS versus CPE filter length under OSNR of 10.5 dB, 12.1 dB, and 15.4 dB for the CPE scheme and the CPE + PAPU scheme. As shown in Figure 5a, Under OSNR of 10.5 dB, 12.1 dB, and 15.4 dB, the CPE scheme exhibits high probability of CS with CPE filter length less than 32, 16, and 8, respectively. As shown in Figure 5b, with OSNR equivalent to 10.5 dB, 12.1 dB, 15.4 dB, to achieve the same PRE-FEC BER (respectively about $2 \times 10^{-2}$, $1 \times 10^{-2}$, $9.6 \times 10^{-4}$), the required CPE filter length of the CPE scheme and the CPE + PAPU scheme are 32 and 8, 16 and 6, and 8 and 8, respectively. With the OSNR equivalent to 10.5 dB, the PRE-FEC BER of CPE scheme sharply degrade from $2 \times 10^{-2}$ to 0.5 with CPE filter length less than 32, moreover, the tendency of PRE-FEC BER agree with the CS probability in Figure 5a. When OSNR is increasing from 12.1 dB to 15.4 dB, the min CPE filter length to avoid the sharply degrading of PRE-FEC BER performance, decreases from 16 to 8.

Figure 6a shows the comparison of PRE-FEC BER and POST-FEC BER versus CPE filter lengths with different OSNR. When the OSNR is 10.5 dB, 12.1 dB, and 15.4 dB, the CPE filter lengths required to achieve $1 \times 10^{-6}$ are 32, 16, 8 for CPE scheme and 20, 6, 2 for CPE + PAPU scheme. As shown in Figure 4, the probability of CS is decreasing and the performance of PRE-FEC BER is improving with the increasing of OSNR, thus the difference between the required CPE filter lengths to achieve $1 \times 10^{-6}$ for two CPE schemes is small with high OSNR. With the lower OSNR, CPE + PAPU scheme is less sensitive to CPE filter length which may reduce the complexity of DSP.

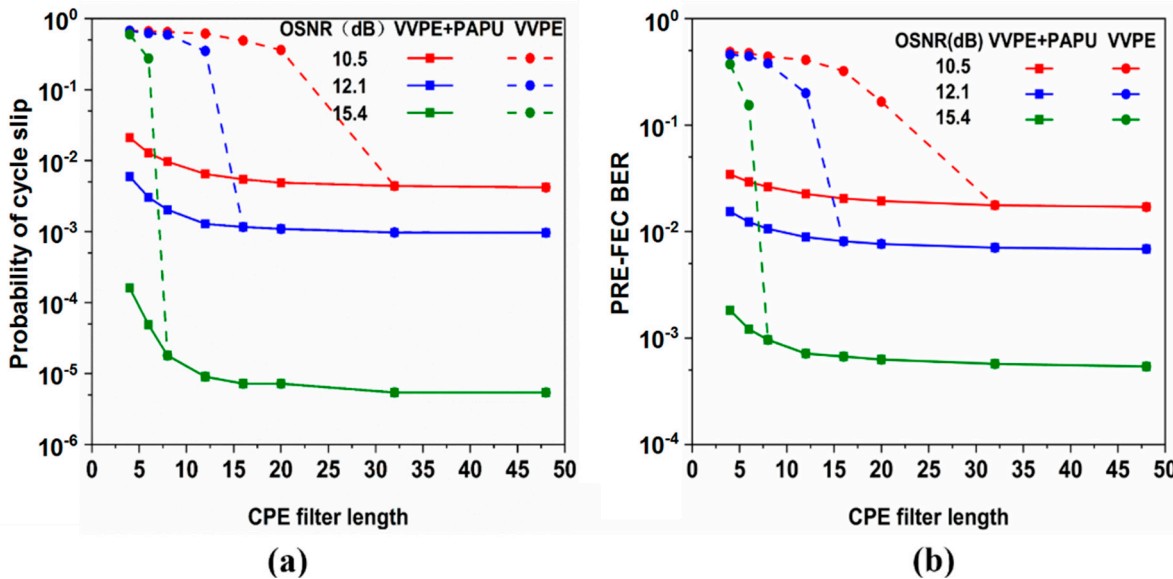

**Figure 5.** (**a**) Probability of cycle slips versus CPE filter length for the CPE scheme and the CPE + PAPU scheme; (**b**) PRE-FEC BER versus CPE filter length for the CPE scheme and the CPE + PAPU scheme, with OSNR equivalent to 10.5 dB, 12.1 dB, and 15.4 dB.

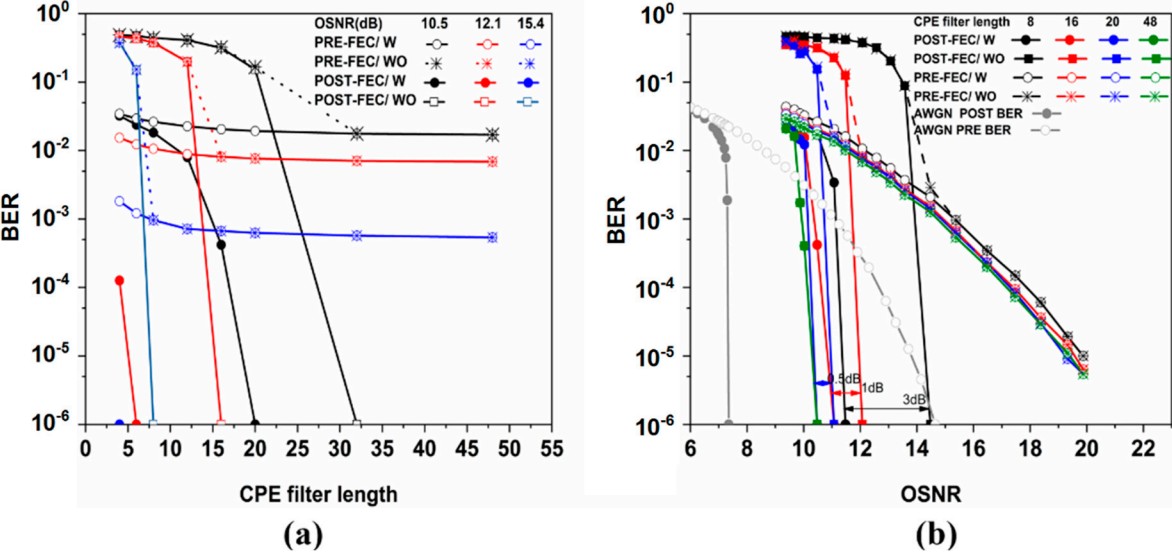

**Figure 6.** (**a**) BER versus CPE filter length curves of CPE and CPE + PAPU, with OSNR equivalent to 10.5 dB, 12.1 dB, and 15.4 dB; (**b**) BER versus OSNR curves of CPE and CPE + PAPU with different CPE filter lengths (W and WO respectively present the CPE +PAPU scheme and the CPE scheme).

The PRE-FEC BER and POST-FEC BER versus OSNR curves under different CPE filter lengths are shown in Figure 6b. Different CPE filter lengths are shown in the label. As is shown, the CPE + PAPU scheme outperforms the CPE scheme by about 3 dB at PRE-FEC BER of $2 \times 10^{-2}$ with CPE filter length less than 16. It can be easily observed that slight degeneration of PRE-FEC BER may lead to rapid deterioration of POST-FEC BER due to CS. Figure 6b also depicts the POST-FEC BER performance of CPE and CPE + PAPU scheme indicated by square and circle connected with solid lines, respectively. The results prove that the CPE + PAPU scheme achieves about 3 dB, 1 dB, and 0.5 dB POST-FEC OSNR gain compared with the CPE scheme under the CPE filter length of 8, 16, and 20. The CPE + PAPU scheme outperforms the CPE scheme under shorter filter length benefitting from its effectively CS mitigation.

Figure 7 shows the relationship between POST-FEC BER and PRE-FEC BER on the CPE + PAPU scheme and the CPE scheme. The AWGN theory curve is represented by dark hollow circle. Apparently, CPE filter length has no significant effect on the BER transfer characteristic for CPE + PAPU scheme. When the CPE filter lengths are respectively equivalent to 8, 16, 20, and 48, the CPE + PAPU scheme successfully decodes signals with $1.4 \times 10^{-2}$, $1.8 \times 10^{-2}$, $1.9 \times 10^{-2}$, and $2.0 \times 10^{-2}$ PRE-FEC BER and the CPE scheme fails to decode signals above $2.5 \times 10^{-3}$, $8.9 \times 10^{-3}$, $1.6 \times 10^{-2}$, and $2.0 \times 10^{-2}$ PRE-FEC BER. As shown in Figure 4a, the CPE + PAPU scheme still present some independent CS, however, these CS can be effectively corrected by FEC. The continuous CS induced by inappropriate PU and amplifier spontaneous-emission (ASE) noise cannot be corrected by FEC. Comparing with the CPE scheme, the CPE + PAPU scheme seems to be less sensitive to CPE filter length. This might be attributed to the fact that the CPE + PAPU scheme effectively mitigates continuous CS induced by phase noise and inappropriate PU.

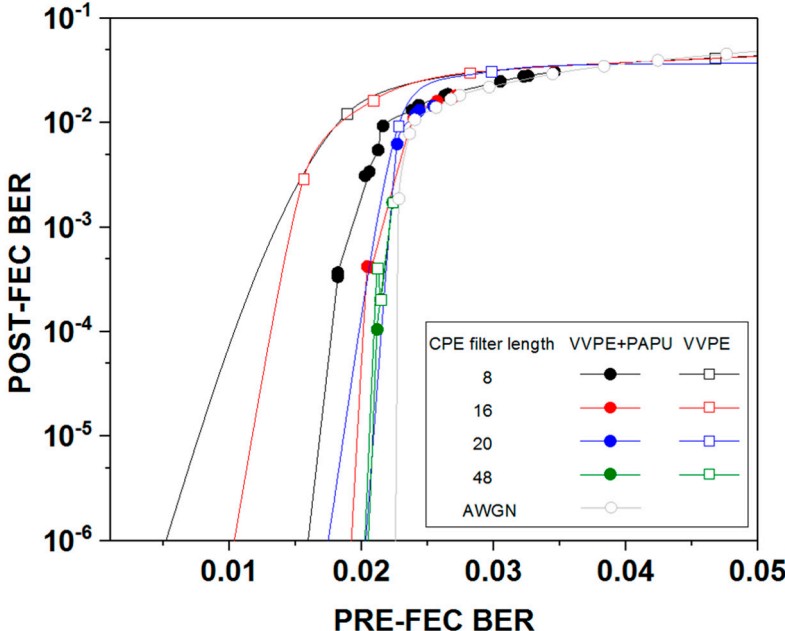

**Figure 7.** The relationship between POST-FEC BER and PRE-FEC BER.

### 4. Conclusions

In this paper, we have experimentally investigated the PRE-FEC BER and POST-FEC BER performance of the CPE + PAPU scheme and the CPE scheme on various CPE filter lengths in 56 Gbit/s QPSK systems. Experimental results show that, compared with CPE, CPE + PAPU can achieve the required PRE-FEC OSNR about 3.1 dB, 1.3 dB, and 0.6 dB at BER of $1.8 \times 10^{-2}$ under the CPE filter length of 8, 16, and 20, meanwhile, the CPE + PAPU scheme respectively obtained 3 dB, 1 dB, and 0.5 dB POST-FEC OSNR gain and a large reduction of the CS rate compared with the CPE scheme, with the CPE filter length of 8, 16, and 20. The experimental results also prove that the CPE + PAPU scheme respectively improve the FEC limit from $2.5 \times 10^{-3}$ to $1.4 \times 10^{-2}$, from $8.9 \times 10^{-3}$ to $1.8 \times 10^{-2}$, and from $1.6 \times 10^{-2}$ to $1.9 \times 10^{-2}$ compared with the CPE scheme, with the CPE filter length of 8, 16, and 20. The experimental results show the CS-rate increases with deterioration of OSNR. On the lower OSNR region, CPE shows high residual CS at short filter length and low residual CS at long filter length, while CPE + PAPU can effectively reduce CS at all the filter lengths. Our experiment results prove that the CPE + PAPU scheme combined with a concatenated soft decision (SD)-LDPC coding can effectively reduce the CS induced by phase noise, which will be a promising candidate for next-generation high-speed coherent optical communications.

**Author Contributions:** This paper was mainly wrote by Y.L. (Yan Li) and Q.N., L.Y. demonstrated the experiment and H.Z. revised the article. C.G., Y.L. (Yuyang Liu), J.Q., W.L. and X.H. contributed to the reviewing and editing of the manuscript. J.W. supervised overall project.

**Funding:** This work was partly funded by 863 program 2015AA015503, NSFC program 61475022, 61505011, 61675034, 61331008, 973 project 2014CB340102, Fund of state key laboratory of IPOC (BUPT), and the fundamental research funds for the central universities.

**Conflicts of Interest:** The authors declare no conflict of interest.

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
