# Peer review of "Post-FEC Performance of Pilot-Aided Carrier Phase Estimation over Cycle Slip"

_applsci, doi:10.3390/app9132749_

Round 1

Reviewer 1 Report

On my remark (the authors to comment on whether a lower BER of 1E-9 or 1E-12 could be achieved ...) which was OPTIONAL anyway, the authors gave a satisfactory answer. I suggest the authors to consider the possibility of including a relevant comment (e.g. in the "Conclusion" section) as a topic for future work. This is also OPTIONAL (as it was my original comment) - the authors do not have to follow it if they do not feel like.

Reviewer 2 Report

The authors experimentally investigated the post-FEC BER performance and the cycle-slip probability of the carrier phase estimation scheme based on Viterbi-Viterbi phase estimation and pilot-aided phase unwrap algorithms. The results and discussions are interesting and helpful for practical application of CPE high speed optical communication systems. This paper can be published in Applied Sciences, provided following issues can be addressed.

Specify and summarize the physical parameters in the optical transmission system in a table.

In Fig. 4(a), the cycle-slip probabilitygoes down to 1e-7. Specify the number of symbols used for this test and explain the reason why the authors could get such a low cycle-slip probabilityvalue.

Supplement some discussions regarding the performance of the Viterbi-Viterbi phase estimation and the pilot-aided phase unwrap algorithms in optical transmission systems with higher-order modulation formats such as 16QAM and 64QAM.

Specify or discuss if the considered transmission system is a back-to-back system or a free-space transmission system. What is the application scenario of this transmission system? 

Round 2

Reviewer 2 Report

The paper can be accepted now.

This manuscript is a resubmission of an earlier submission. The following is a list of the peer review reports and author responses from that submission.

Round 1

Reviewer 1 Report

1.  The paper is too short, lack of theory background, lack of proofs. Before introducing experimental setup and principles, the authors should give more information on literature review with extensively theory analysis.

2.       In the abstract, the research problem is not well presented, idea is not clear enough.

3.       In the paper, the authors experimentally investigate the performance of the CPE+PAPU scheme combined with a concatenated soft decision (SD)-LDPC coding in a 56Gbit/s quadrature phase-shift keying (QPSK) coherent communication system with 62 laser line-width of 300 kHz. The research motivation for this experimentally investigation is not presented in the introduction. Specifically, Why SD-LDPC is selected ? Why 56Gbit/s quadrature phase-shift keying (QPSK) coherent communication system with 62 laser line-width is selected, why 300 kHz ? What is the key idea of this paper compared with previous works of the authors [references 28-30].

4.       Too much reference papers for such a short content.

5.       Quality of all figures (1,2,3,4,5,6,7) are very bad. The authors should improve the quality of all figures as long as at the 100% of the manuscript display, every information in the figure should be easily seen. The author should use vector formats for image e.g.  (encapsulated postscript - .eps).

6.       Color of Figure. 1 is not suitable with the manuscript content.

7.       Figure.7, please make sure in black-white print version of the paper, performance curves and legend could be differentiated.

8.       When the reviewer zoom into Figure 3 (a,b,c,d,e,f) at 400%, the results are not well displayed, therefore there is no meaning for this figure.

9.       Figures 3,4,5,6 should be more commented, please focus on the originality and comparison.

Reviewer 2 Report

Minor changes regarding use of english have to made, e.g. in lines 19 ("release"?) 57 ("A"), 63 ("release"?), 156 ("error free"?), 163-164, 170, 175-176, 179, 195, 196.

OPTIONAL remark: The paper "covers" a BER range of up to 1E-06 which is sufficient for, e.g., telephony applications. However, for more demanding applications, a BER of 1E-9 to 1E-12 may be required. Could the authors comment on whether and how can this BER be achieved? (The authors do NOT have to present more experimental results - a short qualitative comment would be enough).